# Developing a Collaborative Approach to Support Access and Acceptability of Mental Health Care for Refugee Youth: An Exploratory Case Study with Young Afghan Refugees

**DOI:** 10.3390/ijerph21030292

**Published:** 2024-03-01

**Authors:** Jakob Versteele, Cécile Rousseau, Marina Danckaerts, Lucia De Haene

**Affiliations:** 1Faculty of Medicine, University of Leuven, 3000 Leuven, Belgium; 2Paso, University Psychiatric Hospital KU Leuven, 3070 Kortenberg, Belgium; 3Division of Social & Cultural Psychiatry, McGill University, Montréal, QC H3A 0G4, Canada; 4Faculty of Psychology & Educational Sciences, University of Leuven, 3000 Leuven, Belgium; 5Transcultural Trauma Centre for Refugees in PraxisP, Faculty Clinical Centre PraxisP, KU Leuven, 3000 Leuven, Belgium

**Keywords:** unaccompanied refugee minors, unaccompanied asylum-seeking minors, mental health, trauma, access, collaborative care

## Abstract

Despite an increased prevalence of psychiatric morbidity, minor refugees resettled in Western host societies are less likely to access mental health care services than their native peers. This study aims to explore how a collaborative approach can be implemented to promote access to specialized mental health care. Collaborative mental health care embeds specialized intervention in primary care settings and emphasizes the inclusion of minority cultural perspectives through an interdisciplinary, intersectoral network. In this study, we analyze how such a collaborative approach can support access to specialized mental health care for refugee youth. The study presents findings from a qualitative multiple-case study (*n* = 10 refugee patients), conducted in the setting of a psychiatric day program for young refugees that develops an intersectional, collaborative practice in supporting minor refugees’ trajectory from referral to admission. Building on in-depth interviews, participant observation and case documents, within-case analysis and cross-case inductive thematic analysis identify the specific working mechanisms of a collaborative approach. The results indicate how this intersectoral approach addresses the interplay between traumatic suffering and both cultural and structural determinants of mental health. To conclude, a discussion identifies future research directions that may further strengthen the role of collaborative practice in promoting mental health care access for refugee youth.

## 1. Introduction

In 2023, 36.4 million refugees worldwide fled their home countries in search of protection [1]. Refugee children and adolescents make up a considerable group; their life trajectories are marked by pre-migratory stressors of war, atrocities, deprivation and loss or imprisonment of family members, further compounded by cumulative stressors during their flight, including separation from family networks, exploitation and threatening life conditions [2,3]. Once in the host country, they face multiple stressors such as cultural adaptation, residence insecurity, economic distress, social isolation and discrimination [4]. Scholarly work on the mental health sequelae of this accumulation of disruptive life events documents refugee minors as an at-risk population, with a significantly increased prevalence rate of posttraumatic stress disorder (PTSD) and major depressive disorder (MDD) [5,6,7]. Unaccompanied refugee minors (URMs) make up a particularly vulnerable group, as they live through these numerous hardships without the immediate support and protection of parents or guardians. Research shows that URMs experience a disproportionately high trauma load, including physical violence as a very common flight experience and sexual violence and forced family separation reported by at least one-third of URMs [2]. Furthermore, a growing body of evidence documents associations between levels of posttraumatic stress symptoms (PTSSs) and low-support living arrangements, refusal or insecurity of resident status, perceived discrimination and daily stressors experienced by minor refugees [8,9,10]. Residing in low-support living arrangements leads to a significantly higher prevalence of PTSD [11], while low support during the asylum process and uncertainty or refusal of asylum seem to negatively impact levels of PTSD and the overall mental health trajectory [12,13,14]. Studies have consistently demonstrated that prevalence rates of mental distress and psychiatric disorders for URMs are more elevated than for accompanied refugee minors [10,15,16].

### 1.1. Barriers to Mental Health Care for URMs

In spite of this elevated morbidity risk, studies consistently document how young refugees have fewer contacts with mental health services than host country youth [17,18,19,20]. This underutilization is explained by the complex interplay of barriers to access and engagement in mental health services for refugee communities [21,22,23].

Addressing these barriers and related health disparities requires attention to the cultural differences and social inequities that inform them [24,25]. In order to enable services and clinicians to acknowledge and act upon these barriers, clinical literature emphasizes the need to enhance professionals’ cultural and structural competence [26].

### 1.2. Central Perspectives in Promoting Access: Cultural Competence, Structural Competence and Cultural Safety

First, promoting clinicians’ cultural competence entails strengthening knowledge on social and cultural influences on patients’ health beliefs and behaviors, as well as the skills to responsively work with those factors [27]. A lack of cultural competence may explain barriers to access to mental health care in several ways. For example, patients may use specific idioms of distress and explanatory models to express and understand their suffering while not recognizing how these may warrant mental health care, and service providers may similarly fail to identify these idioms of distress [23,28]. Next, stigma within refugee communities provokes significant thresholds to mental health care. Within refugee communities, psychological or psychiatric symptoms are frequently associated with fears of social isolation, incarceration or institutionalization, and the consequences associated with help-seeking are often considered to be more damaging than receiving no care at all [29,30]. Moreover, cultural competence as a means to promote access should prevent a lack of understanding of how cultural notions of self and community can lead to prioritizing academic trajectories or familial responsibilities or a preference for self-help or informal [23]. However, creating such an understanding of cultural factors in relation to URMs might be hampered due to the fact that separation from family and country of origin causes these cultural narratives to evoke a sense of loss, discontinuity and loneliness [31,32,33].

Second, the emphasis on promoting clinicians’ cultural competence in removing barriers to mental health care has been broadened by a growing emphasis on structural competence. This clinical framework is focused on clinicians’ responsibility to address structural determinants of health and health inequalities [34,35]. In access to mental health care for refugee communities, several structural barriers are at play, such as complex health care systems, waiting lists, eligibility criteria, high costs and the lack of cultural brokers or interpreters limiting the capacity of parents and youth to obtain appropriate care. Furthermore, the growing hostility towards refugees in host countries’ social and policy environments [36] not only predicts mental health problems [37,38] but also encroaches upon mental health services in myriad ways. Institutional racism, when biases or discrimination become ingrained in organizational and regulatory practices, may lead to clinical encounters in which deficit perspectives on cultural differences inform micro-interactions that reiterate prejudice, experiences of marginalization and retaliation [39]. Research shows this leads to increased mistrust in institutions, decreased efficacy of evidence-based mental health treatments [37,40,41] and further increased barriers to access to healthcare [42,43]. The combining of cultural and structural competence attempts to answer the systemic limitations of cultural competence in adapting healthcare encounters to individuals and communities of different socio-cultural backgrounds and encourages clinicians to also explicitly direct attention towards the historical and ongoing social factors that make that clinical encounter unsafe. This is in line with guidelines on equality, diversity and inclusion (EDI) which increasingly call for cultural safety [44]. Addressing the many levels of direct, covert and structural oppression may help to avoid individualized and pathologized interpretations of perceived avoidance, anger and distrust and may help to understand them as strategies to survive ongoing discrimination [45].

### 1.3. Refining Cultural and Structural Competence in Working with Refugees: Trauma-Informed Practice

Although practices rooted within the promotion of professionals’ cultural and structural competence provide an array of strategies for promoting access, ensuring access to mental health care for refugees additionally requires an approach that addresses how patients’ trauma responses and specific social predicaments may intersect with cultural and structural factors. With only a paucity of scholarly work addressing the interaction between refugees’ traumatic suffering and cultural and structural health factors in accessing mental health care [46,47,48,49,50], an in-depth exploration of how these factors are marked by traumatic life experiences seems particularly relevant and may point to ways in which strategies of cultural and structural competence can engage with the specificities of working with trauma in promoting access to mental health care for refugees.

Several dimensions of traumatic suffering may intersect with cultural and structural factors. First, distrust that stems from trauma may affect the development and dynamics of the clinical relationship. Second, traumatic stress can cause dysregulation and re-enactment in that clinical relationship, and third, trauma often comes with a certain complexity in negotiating disclosure. Research shows that past experiences of young refugees impact their levels of interpersonal and institutional trust which in turn may reduce their ability to engage with mental health care services [51]. The life histories of refugees frequently include abuse of power by those in authority. The consequent institutional distrust impacts asylum seekers and refugees in their ability to determine whether or not agencies are independent of the state [48,52]. The context of prolonged asylum procedures and their focus on truth-telling form a social context in which mistrust becomes a prevalent issue among refugees [53]. When a mental health encounter takes place, it can be characterized by the avoidance of topics, as well as by the risk of re-traumatization through the repetitive disclosure of a traumatic past. Continuous renegotiating of disclosure is therefore an important aspect of each clinical encounter [49]. Creating and managing a “holding environment”, in which the intensity of traumatic experiences can be endured and transference and counter-transference and traumatic re-enactment are managed, can, however, be a challenging endeavor [49,54].

### 1.4. Collaborative Mental Health Care: Intersectoral Work in Promoting Mental Health Care Access

Collaborative mental health care (CMHC) [55,56] is a model of psychosocial care developed in Canada as a systemic approach to enable access to marginalized communities through embedding specialized mental health care professionals in primary care settings. These specialized mental health consultants coordinate collaborative, interdisciplinary, intersectoral and intercultural networks surrounding children and family, actively including patients’ minority voices. The intersectoral approach connects mental health diagnosis and treatment to targeting social and structural determinants of illness. Existing evidence on CMHC shows how it can improve access by decreasing stigma and fostering alignment on cultural factors in patients’ illness experience, creating higher levels of satisfaction and therapeutic adherence [57]. Furthermore, studies indicate how a collaborative approach allows structural determinants of mental health to be targeted [58]. While studies on collaborative mental health care have included refugee communities, no studies as of yet have explicitly explored how a collaborative approach can address the intersection of cultural and structural factors with refugee patients’ traumatic suffering in promoting access to care. Furthermore, studies on CMHC with URMs are lacking. It is important to note however that collaborative mental health care is a systemic approach that focuses on enabling a dialogue in which adolescents, parents and professionals have an equal voice [58,59]. The involvement of parents in the trajectory of URMs is of course significantly impeded by the still ongoing separation, potentially impacting the applicability of CMHC as an approach for the URM population.

### 1.5. Implementing CMHC in Promoting Access to Mental Health Care for URMs: A Pilot Study

This study presents an intervention description of a collaborative approach aimed at promoting access to residential mental health care for URMs. For this purpose, we collected primary-level evidence in accordance with Veerman and Van Yperen’s model for developing evidence-based practice [60] and propose to answer the following research question: “How does a collaborative approach to promoting access to mental health care for URMs allow for working with the interplay between cultural and structural determinants of mental health and traumatic suffering?”

## 2. Materials and Methods

### 2.1. Method

This study aims to describe an outpatient program that was set up to create a care pathway towards a newly opened child psychiatric day care unit, Paso, using the principles of CMHC. The descriptive nature of this study and the proposed theoretical underpinning of this intervention aimed at promoting access to mental health care is in accordance with Levels 1 and 2 of Veerman and Van Yperen’s model for practice-based effectivity research [60]. Level 1 is descriptive levels of evidence detailing the essential elements of the intervention, and Level 2 is theoretical levels of evidence detailing a rationale based on existing literature. An exploratory multiple-case study was undertaken to enable a contextualized, “thick description”, that offers a direct connection to cultural theory and scientific knowledge [61,62]. This format enables in-depth exploration of complex phenomena in real-life situations [63]. Multiple-case study designs, analyzing more than one case study in parallel, allow comparisons across several cases and are therefore considered to be more robust than a single-case study [64]. Gathering evidence from multiple cases can enable the generalizability of the findings and allow the development of theory [65].

### 2.2. Setting

Paso is a novel unit for minor refugees at UPC KU Leuven located in Kortenberg, Belgium, developed and coordinated together with the Transcultural Refugee Trauma Care Service at PraxisP, Faculty Clinical Centre of the Faculty of Psychology & Educational Sciences, KU Leuven. It is a child psychiatric day hospital for refugee minors that provides a multimodal, transcultural trauma treatment program aimed at stabilization, integration and reconnection, with a gradually shifting focus from supporting symptom reduction, affect regulation and adaptive coping towards trauma narration and an active, meaningful re-engagement with developmental tasks and future perspectives [66,67]. During their time at the day clinic, the URMs reside in 3 group homes of residential youth care organizations located in the near environment of the day clinic. This entails that choosing to participate in the treatment program also means temporarily moving to one of the group homes.

To facilitate access to the day clinic, a collaborative, ambulatory program was set up that is aimed at exploring illness experience and overcoming barriers towards mental health care and throughout which a therapeutic mandate is negotiated with the URMs and the network partners surrounding them. Team members of the day clinic involved in this intake process have different professional backgrounds. The lead in this intervention is taken by a clinical psychologist of the day clinic, often accompanied by the cultural mediator of the day clinic; the trajectory is supervised by a child psychiatrist of the day clinic who also is present in one consultation. A representative of the future group home also takes part in one meeting. All meetings took place in the presence of an interpreter. The network partners involved in the referral have various professional roles: legal guardians, counselors of reception centers or group homes, ambulatory therapists, representatives of the school, etc. The program consists of an initial meeting with the network partners to discuss the case, explain the collaborative intervention and explore possible pathways for collaboratively negotiating access and the therapeutic mandate. In this meeting, the potential roles and positions of both the network partners and the team members of the day clinic and the group homes throughout this process are discussed. This is followed by a minimum of three meetings with the network partners and the URM that take place in varying locations (the reception center or group home where the URM resides, the day clinic and the group home where the URM will reside during admission in the day clinic). In these meetings of approximately one hour, the day clinic team members explore how the referral to semi-residential mental health care is experienced by the URM, which opportunities are seen in this referral and which barriers seem to make semi-residential care less indicated. Throughout this exploration, opportunities are created to collaboratively construct shared therapeutic goals that allow for admission in the semi-residential psychiatric treatment program to become acceptable and appropriate.

### 2.3. Cases

For this study, ten cases in which URMs completed the entire intervention as it was originally designed were included. Cases in which the intervention was aborted before completion were not included, given that the intention of this study was to provide a description of the entire intervention. Interventional trajectories for the recruited cases took place during a period spanning from January 2021 through May 2022. During this time span, 20 interventional trajectories took place. In these 20 trajectories, there were 4 cases in which the URM did not complete the entire intervention. As it turned out, all of the URMs participating were Pashto-speaking, Afghan adolescents aged 15–17, with an average age of 16.1 years old. They were in various stages of Dutch language acquisition, but none of them had mastered the Dutch language to the extent that they would be able to have a mental health encounter in Dutch. All of them were still in various stages of their asylum procedure and were residing in reception centers or smaller group homes that are part of the national reception network. They all showed various symptoms that could be attributed to traumatic stress and warranted semi-residential psychiatric treatment. Symptoms included the following: somatoform complaints, psychogenic non-epileptic seizures (PNESs), dissociative episodes, difficulties in emotional and behavioral regulation, depressive mood, anxiety, insomnia, nightmares, loss of appetite, suicidal ideation and others. An overview of the cases that were included in this study can be found in Table 1.

### 2.4. Data Collection

Multiple types of qualitative data were collected. We made use of case documents containing detailed notes from the actual intake meetings as well as notes from intermittent process reflections and of an intervision process set up among the clinical team members to develop this intervention and several qualitative interviews exploring the experiences of participants in the intake process. Qualitative interviews were conducted with four team members of the day clinic, one URM and one legal guardian who was involved in two of the cases. An overview of the participants who were interviewed can be found in Table 2.

### 2.5. Data Analysis

In the first phase, a within-case analysis was conducted on the data concerning individual intervention trajectories. Interventional processes for overcoming cultural and structural barriers within each individual case were detailed and structured. The clinical documents available and the transcript of the URM interview were examined by means of a thematic narrative analysis [68] through a close contextualized reading. Subsequently, the identified collaborative processes were brought together in a restructured dataset. In a similar fashion, interviews with people other than the URM involved in the intake process and documents of the intervision process were analyzed by a close contextualized reading, and the identified barriers and interventional processes were added to the restructured dataset. On this restructured dataset, two researchers independently conducted a cross-case inductive thematic analysis [69]. Initial codes were reviewed by rereading coded material in order to track relationships between codes, prominent overarching themes and subthemes relevant to our research question. After identifying relevant themes and subthemes, cross-case themes concerning the use of principles of CMCH to facilitate access and initiate a therapeutic mandate were defined, reviewed and refined.

## 3. Results

In analyzing the aforementioned data, we found that the intake procedure in these cases consisted of one preparatory contact with referring partners followed by three to five intake meetings with the URM present. The average amount of intake meetings was 3.6 with the URM present and 1 without the URM. Thematic analysis generated six themes, identified as central to the collaborative process aimed at mediating barriers, facilitating access and initiating a therapeutic mandate. The themes are ordered according to the extent to which they occurred in the analyzed documents and are presented in a way that attempts to reveal both the content as well as the collaborative process within each theme. Each theme is divided into several subthemes. An overview of all themes and subthemes is presented in Table 3.

### 3.1. Building the Capacity to Understand and Space to Contain Symptoms of Traumatic Stress 

A central cross-case theme pertained to the role of the intervention in working with traumatic stress. This trauma-informed approach aims to both promote relational security as well as explore relevant mental health needs. In doing so, it creates opportunities for the promotion of expertise in working with traumatic stress in network partners.

#### 3.1.1. Creating a Holding Environment

First of all, we found that a considerable part of the intake process focuses on creating a “holding environment” for symptoms of traumatic stress. Symptoms of traumatic stress fluctuate over time during the intake process. They may worsen, or symptoms may temporarily go into remission. As a result, needs may become less apparent, or ruptures between URMs and caregivers or among caregivers themselves can appear when symptomatology worsens. Throughout the intake process, clinical team members focus on how to deal with these fluctuations and how to prevent them from affecting the commitment of the network partners or the URMs. To this extent, the clinical team members provide psychoeducation on how to understand these fluctuations as part of the symptomatology of traumatic stress, and subsequently, a plan is made collaboratively with the network partners at the start of the intake procedure to manage these fluctuations and on their impact on commitment.

The intervision documents detail an important focus on the first meeting with the network partners prior to the intake process. The goals of this first meeting are forming an idea of collaboration, expressed by clinicians as “*We are going to do this together.*”, anticipating potential splitting between partners due to traumatic symptomatology, anticipating how to retain a perspective of understanding and care when the intensity of behavioral symptomatology makes this difficult.

In the case documents of URM 9, it is detailed how clinicians propose clear roles between network partners to allow room for differences in communication style towards the URM. This is in an attempt to enable all partners to maintain a position of care towards the URM in case intense behavioral symptomatology makes it difficult to stick to a care perspective.

#### 3.1.2. Collaboratively Exploring Symptoms of Traumatic Stress

Second, our analysis showed that during the intake procedure, a thorough exploration of trauma symptoms takes place in which both the URMs and the network partners act as informants and the clinical team members act as observers and inquirers. Conversation and meaning are created around these trauma symptoms. The clinicians enquire about symptoms of traumatic stress and encourage network partners to share their own observations and to support the URMs in identifying possible symptoms. These symptoms are linked to past life experiences in order to understand and label them both as a means of survival in difficult circumstances and as a need for future care. This creates opportunities for all partners to clearly voice their intention of care in correspondence to those needs, and consequently to connect these needs to the proposed psychiatric treatment, thus creating important stepping stones towards a shared request for help and consent to admission.

*“I doubted about coming to the day clinic because of the trees surrounding it. They brought back many unsettling memories and I was afraid that going to the day clinic would destabilize me because of this. That forest made me think of my migration route, I must have fainted 3 times after visiting it because of how stressful it was.”* In the intake meeting after his first visit to the day hospital, URM 1 explains how activation of symptoms of traumatic stress created doubt, but also how the intake conversations helped him view psychiatric care as possibly appropriate. This experience helped him understand how the treatment program might be helpful for him.

#### 3.1.3. Managing Traumatic Symptoms during the Intake Meetings

Additionally, our results show that when URMs avoid talking about their psychological and/or psychiatric suffering due to stigma, posttraumatic avoidance or silencing, the network partners are asked to function as facilitators that either voice the URMs’ symptomatology or help the URMs in voicing their symptoms. Simultaneously, the clinical team members voice symptomatology that may impede the intake process: avoidance, prodromal signs of dissociation, heightened levels of arousal, etc. This creates the possibility of educating the referring team and creates awareness for the URMs. It also offers opportunities to the network to offer co-regulation or suggest working coping strategies and allow a URM to be an agent in explaining what would be helpful in staying regulated.

In the case of URM 5, it was observed that the URM at times seemed submissive and was often overwhelmed by severe, paroxysmal headaches and intense feelings of sadness. This made reciprocal conversation difficult. Several conversations with network partners took place on how to understand and voice this symptomatology and on how each member of the network could help support the URM in these moments.

#### 3.1.4. Collaboratively Dealing with Traumatic Re-Enactment

The clinical team members spend time modeling the capacity to understand and take in negative projections and traumatic experiences that are being transmitted in order to avoid ruptures in the intake process. Within a network that is being formed, relational and institutional re-enactment of trauma may take place, putting pressure on the alliance that is being formed, possibly resulting in splitting in between professionals and between professionals and the URMs. By explaining possible relational dynamics and modeling the capacity to understand and contain them, network partners are supported in dealing with this.

In the case of URM 9, polarized views on how to understand the behavioral symptoms of the URM along a “victim–perpetrator continuum” risked leading to a rupture between the legal guardian and the care professionals of the group home where the URM was residing. In this case, creating a shared understanding of the issues at hand, connecting the concerns of both the guardian and the group home, renewed the idea of a network surrounding the URM that aimed to facilitate access to the day hospital.

### 3.2. Shedding Light on Structural Factors Impacting the Experience of URMs and Collaboratively Modeling and Initiating Relative Repair

Our findings showcase how the impact of structural factors on URMs’ suffering was apparent throughout the 10 cases and throughout the intake trajectory. The collaborative intervention aims to acknowledge these factors, avoid re-traumatization through coercive dynamics and move beyond a mere medicalization of the suffering by attending to structural determinants as predictors of mental health. This process simultaneously allows for the enhancement of structural competence in collaborative partners.

#### 3.2.1. Addressing and Containing the Impact of Migration, Asylum and Reception Policies

Through our analysis, several conversational topics concerning structural factors that may impede access were identified. First of all, many of these topics, revealed both by URMs and network partners, reflect how policies concerning immigration and refugee reception can create a hostile environment for URMs that affects the trust and safety needed for young refugees to perceive care as safe and acceptable.

Case documents show how the insecurity surrounding his asylum procedure created visibly heightened levels of arousal in URM 3 throughout the intake procedure. He often directed the conversation towards questions and demands concerning his procedure, overshadowing the conversation about appropriate mental health care.

Case documents show that the directors of the reception center where URM 8 resides state that if the intake procedure is not successful, they will likely give the URM a definitive disciplinary transfer to another reception center.

Analysis shows that the clinical team actively tries to elicit these narratives throughout the intervention trajectories and specifically puts emphasis on the potential impact of these policies concerning both immigration (violent border interactions, lengthy and stressful asylum procedures, etc.) and reception (crowded and understaffed reception centers, repetitive time-outs, disciplinary sanctions and transfers, etc.) on the suffering of the URMs and their perception of care. In response to this, the clinical team members express their understanding of how these factors contribute to the URMs’ suffering and create room for their symptomatology to be understood as an expression of these structural factors. In parallel, this allows for their resistance against hospitalization to be understood as a resistance against the psychologization of suffering that is determined by these structural factors. Network partners are simultaneously invited to express their understanding of how these factors contribute to the suffering of the URMs.

#### 3.2.2. Altering Coercive and Authoritarian Dynamics within the Collaborative Network

Secondly, we found that possible coercive and authoritarian dynamics within the collaborative network potentially mirror the aforementioned policies. These dynamics are explored by the clinical team and discussed in a continuous conversation with the network partners. This is done in an effort to better understand the worries they may reflect. Within the same conversation, solutions are discussed to move away from coercion towards a caring yet directive approach that enables the expression of underlying concerns. This is done in an attempt to avoid re-traumatization due to repeating past experiences of coercion and feelings of powerlessness and to move towards a more agent position for the URMs.

The guardian of URM 9 uses an authoritarian approach in response to the URM’s resistance to the day hospital: “*I am the guardian, the adult, I set the rules. He will have no choice but to follow this. I will no longer be embarrassed*”. The case documents show that a further inquiry into the meaning of this outcry leads to the guardian opening up about how stressed he is about his responsibilities towards the URM and the developmental tasks set for him in resettlement: asylum procedure, integrating into society, building a successful future for himself.

#### 3.2.3. Collaboratively Dealing with Institutional Distrust

Third, it was found that the impact of these structural factors on institutional distrust is explored and acknowledged. Consequently, the clinical team attempts to repair this institutional by putting a strong focus on voluntary care, transparency and professional secrecy. In this process, agency in the URMs’ decision making, is collaboratively encouraged and consent is carefully negotiated with all involved partners.

“*Is the day clinic a government program? Do I have to come because the government says so? Because that’s what my lawyer told me.*” When asked about his outlook on a possible admission, URM 9 asked these questions, demonstrating how the authoritarian approach in parts of his network interacts with institutional distrust. In response to this, the clinical team acknowledges the confusion or suspicion and at the same time repeats that admission in the day hospital is voluntary and will only happen if the URM actively consents to it.

“*This place looks like a prison, what if you’ll poison me through the food? In every land I’ve been, I’ve been beaten up by police. In Europe they lock you up, if you misbehave, or they give you “an injection”.*” When exploring possible reasons for admission, URM 2 explains his resistance to admission and general distrust against the backdrop of his experience of how refugees are treated in Europe, echoing a hostile migration policy. In response to these concerns, the collaborative network expresses their disapproval of those practices and explains that admission in the day hospital is voluntary and requires the active consent of the URMs.

#### 3.2.4. Constructing a Shared Therapeutic Goal

Finally, our findings show the intake process focuses on collaboratively creating a shared therapeutic goal. An important focus is put on the encouragement of an agent position for the URM and a reduced level of coercion. Divergent positions within the network on the therapeutic goals are considered. Network partners are supported in expressing their worries informing their referral with the care and directivity reminiscent of the position of a primary caregiver. The clinical team repeatedly explains the importance of consent in coming to a shared therapeutic goal in order to effectively work around these worries. Similarly, the URM is encouraged to express personal therapeutic goals and is supported in this by the network partners expressing potential therapeutic goals and the clinical team who explain therapeutic possibilities at the day hospital. In parallel, appropriate strategies to effectively negotiate consent are discussed with network partners to collaboratively move towards convergence on shared therapeutic goals. In the formation of shared therapeutic goals, the worries, cares and opinions of absent primary caregivers and the extended family network are explored as potential therapeutic goals. This allows URMs to align their agent position with the concerns of the network partners and the ideas and values of their family network.

### 3.3. Exploring and Acknowledging Cultural Factors, Allowing Cultural Continuity and Modeling Cultural Sensitivity

Throughout the intake process, cultural differences both in illness experience and in other life domains are actively addressed and explored in order to develop a working relationship that seeks to welcome these differences as essential in informing the delivery of culturally safe mental health care for URMs as well as promote continuity in URMs’ trajectories in exile.

#### 3.3.1. Welcoming Cultural Idioms of Distress, Explanatory Models and Coping Strategies

Our analysis revealed conversations concerning the important role of cultural factors in illness experience in which both the URMs and the network partners act as informants. From the start, culturally informed symptom expressions, explanatory models and healing practices are explored and acknowledged. Network partners also join in the acknowledgment of the significance of these influences.

The guardian of URM 4 mentions how he learned that his pupil considers the impact of the evil eye important. When this is mentioned, the URM explains how he received a protective amulet from his uncle, but how he lost it. He then goes on to talk about how a very close and important friend of his made him a new amulet.

URM 2 mentioned how his troubles started in Afghanistan and how he, his surrounding family network and the religious figures they consulted considered his complaints due to possession by djinns. He explains how he was offered specific healing practices by the local Imam and says his family believes this will help him. He describes how he adheres to this healing practice and explains how this is also a way of being loyal to his family and their understanding of his illness.

Throughout these conversations, cultural divergence in explanatory models and healing practices is proposed and explained. Within this dialogue, the day hospital is explored as potentially appropriate care. This exploration creates opportunities for both clinical team members and the network partners to explain how the referral to the day hospital encompasses a sensitivity towards and respect for these cultural differences.

#### 3.3.2. Modeling Cultural Continuity

Meaningful cultural experiences separate from the illness experience are explored and acknowledged, and cultural continuity is modeled by imagining space for sports, cuisine and religion within the care program.

During his first intake conversation, URM 1 is watching other young Afghans playing cricket. The clinical team member tells him she notices how he seems to find enjoyment in watching the other Afghans play cricket. He mentions how cricket is very important to him. His eyes light up when in a later intake meeting, the future group home showcases a running team and cricket opportunities. The person responsible for the group home also discusses cooking possibilities and mentions how he looks forward to eating “Afghan rice” again now that there will be an Afghan adolescent staying with them. This visibly creates a connection with the young refugee and seems to put him at ease.

A space for cultural continuity is created by modeling cultural diversity within the intake procedure by the presence of interpreters, intercultural mediators and team members with a diverse background. “URM 2” talks about this in his interview, saying the following:

“*When there’s a translator present, it feels like I’m home, in Afghanistan. Therefore I see translators like a brother or a sister.*”

### 3.4. Moderating a Multi-Voiced Conversation That Explores, Gives Meaning to and Counteracts Stigma

In an attempt to counteract stigma, the intervention redefines the potential meanings of psychiatric care by modeling mental health care as socially embedded care that works towards the realization of personal goals and social integration.

#### 3.4.1. Exploring the Meaning of Stigma

First of all, we found that stigma associated with mental health problems and help-seeking in mental health services was mentioned as a recurring barrier for URMs in considering the day hospital as appropriate care. Our analysis showed that clinical team members systematically explore stigma by asking questions about this topic, but also by answering “unasked” questions to open up conversation about this theme. This allows the URMs to recognize stigma if present and talk about their concerns. Several team members mention in their interviews how in their experience stigma could be traced back to associations between mental health problems and psychiatric and or neurological disorders such as schizophrenia, severe neurodevelopmental or neurodegenerative disorders, epilepsy and substance abuse and the social isolation people with these disorders experience in URMs’ home countries. These notions seem to interact with the URMs’ personal experiences of social isolation. According to the interviews of team members, URMs spoke about their fear that accepting psychiatric care would almost automatically lead to them being part of a group of people who are considered “crazy”, who have no control over themselves and for whom there is no cure. The team members explain that acknowledging this fear felt like an important first step in dealing with this barrier.

#### 3.4.2. Establishing Trust

Second, it was found that network partners who already had a longer standing and hopefully more trusting relationship are well positioned to help create a trustworthy and safe enough setting in which stigma and the underlying fears can be discussed. Network partners are positioned as informants who bridge the gap between the clinical team and the URMs by explicitly asking them to explain their understanding of the symptomatology and suffering of the URMs.

#### 3.4.3. Redefining Psychiatric Symptomatology and Psychiatric Care

Finally, several cases demonstrated that acknowledging symptoms of traumatic stress as protective mechanisms and actively normalizing symptoms as a means of survival in the face of extreme life stressors by both the clinical team members and the network partners may ease some of the fears rooted in stigma. Simultaneously, efforts are made to ease potential fears of loss of social position or exclusion. The proposal of socially embedded psychiatric care is made possible by the multidisciplinary composition of the network. The different disciplinary positions allow for a credible presentation of how the proposed care aims to be socially embedded. Psychiatric treatment is connected to a hospital school trajectory aimed at language acquisition and a group home where URMs are supported in finding independence, student jobs or suitable leisure activities. This connects psychiatric treatment to goals beyond symptom reduction and the missions URMs carry with them (e.g., family reunification, learning the language of the host country, securing a future through schooling), allowing the construction of alternative meanings for psychiatric care.

### 3.5. Responding to Multiple Ruptures and Experiences of Loss by Imagining and Enabling Continuity

The collaborative approach works towards establishing relational continuity through the continued involvement of referral partners, but also by explicitly inviting primary caregivers and extended family networks to be a part of the exploration.

#### 3.5.1. Exploring Feelings of Rupture and Loss

Our research showed that several intervention trajectories bore witness to multiple ruptures and experiences of loss that impact the mental health status of the URMs. In this regard, clinical team members explore feelings of loss and rupture that may be linked to the move to the day hospital and a new group home. At the same time, the continued involvement of network partners throughout the URMs’ stay at the day hospital is negotiated to enable a sense of continuity and lessen the feelings of loss associated with admission to the day hospital.

#### 3.5.2. Supportive Positioning of Network Partners

We found that network partners are positioned as supporters who help the URMs make appropriate decisions in the absence of their primary caregivers and extended family network. Case documents show how this was aptly voiced by some of the URMs:

“*I don’t have any parents here, who can help me make this decision, I’m going to need your help to make it with/for me.*” URM 5 explicitly asked the network partners for help in deciding whether admission would be a good idea.

However, the combination of being both referring professionals and supporters in this process of decision making is a complex position that can easily be interpreted by the URMs as dishonest, leading to a sense of being rejected rather than being supported.

“*You’d rather see me leave.*” This was the initial response of URM 2 when care partners voiced their beliefs that residential mental health care would be an appropriate place for him.

#### 3.5.3. Establishing Continuity of Family Relations

Alongside this supportive positioning, the intake process enquired about what primary caregivers, the extended family network or important peers would think about admission. In this way, the continuity of relation with family and broader network figures was imagined. This is exemplified in the case documents of “URM9”:

“*I asked my friends and they said that if I think this will help me, they would support me.*” URM 9 was asked what his family or friends would think about admission. Initially, he was unable to answer this question, but at the next meeting, he explained that he had received their support.

In line with this theme, inquiries are made about filial duty and mission to find ways to connect these aspects to the reasons for referral, thus creating an opportunity for the URMs to align the proposed care with maintaining their position within their families. In this network, partners act both as informants and as supporters.

URM 2 mentions in his interview how he felt his suicidal ideations conflicted with his filial responsibilities (e.g., academic expectations, family reunification) and explained how this helped him decide admission to the day hospital was a good choice to make.

### 3.6. Forming a Network Surrounding the URMs to Counteract Loneliness and Isolation by Embedding Care in Society

Surrounding the URMs with a network of people invested in providing appropriate care makes it possible to propose psychiatric treatment as a form of collective, community-oriented care that intends to counteract feelings of loneliness and isolation and to alleviate the impact of significant life experiences that inform those feelings.

#### 3.6.1. Exploring Feelings of Loneliness and Isolation

First of all, several team members mention in their interviews how isolated many of the encountered URMs are, and case documents of several URMs document how they expressed a sense of loneliness that impacts their mental health.

“*My only problem is being apart from my family. I miss my family and I’m lonely.*” When URM 8 was asked if he felt there were any problems that an admission could help him with, he explained the sense of loneliness that he felt after family separation.

Case documents show how experiences of loneliness and isolation due to the separation from family, including dimensions of survivor’s guilt, are explored and acknowledged.

#### 3.6.2. Surrounding URMs with a Network

Secondly, we found that attempts are made to counteract this feeling of loneliness and isolation by formally surrounding URMs with a network of partners as a representation of social and collective care. In forming this, network family relations are actively represented in the conversation to visualize their inclusion as partners surrounding the URMs.

“*I didn’t have my family with me, but at times it felt as if they were kind of a family for me. It’s not the same, but they really wanted me to feel good.*” URM 2 speaks in his interview on what he found helpful in dealing with his initial fear of admission.

#### 3.6.3. Envisioning Socially Embedded Care

Lastly, we found that team members of the group homes are introduced as possible collaborative partners to show how the care program consisting of both a day hospital and a group home setting offers socially embedded care that moves beyond a focus on solely symptom reduction. The presence of team members of the group homes within the intake trajectory allows for a tangible explanation of how psychiatric treatment does not have to mean social exclusion and how social embedding can be both a means to achieve symptom reduction as well as a goal enabled by symptom reduction. Counselors of the group home setting do this by showing how they house URMs with and without apparent psychiatric needs and how they support all of the URMs in daily needs. They explain how the group home setting and the day clinic team will support the URMs in finding suitable living arrangements after discharge from the day hospital.

## 4. Discussion

In this article, we aimed to present a thick description of an intervention based on principles of collaborative care that was set up to create a care pathway towards an adolescent psychiatric day care unit for the transcultural trauma treatment of minor refugees. We analyzed case documents of 10 collaborative interventions together with intervision notes and interviews with both URMs and professional participants. Our analysis focused on the question “How does a collaborative approach to promoting access to mental health care for URMs allow for working with the interplay between cultural, structural determinants of mental health and traumatic suffering?” and led to the identification of six themes that illustrate both the content and the collaborative process and are central to the intervention.

We intend to look more specifically at how the proposed collaborative approach may potentially facilitate addressing the specific confluence of potential barriers stemming from cultural and structural determinants of mental health and traumatic suffering. Relational distrust provoked by trauma and institutional violence may affect the development of a clinical relationship. Furthermore, conversations about both cultural and structural hampering factors can lead to the re-activation of trauma due to the way these factors may be imbued by traumatic meaning. Building on our findings, this discussion aims to further elaborate on the potential underpinning mechanisms of collaborative approaches to access promotion in working with these trauma-related, cultural and structural dimensions in URMs’ functioning.

First of all, the intervention seeks to establish a “safe enough” sense of connection between the URMs, the collaborative partners and the members of the clinical team that can potentially enable the imagination of “safe enough” attachment [45]. This is done with the reality in mind that trauma may have impacted the URMs’ fundamental assumptions about the safety of the world [70]. In order to develop this “safe enough” sense of connection, the clinical members of the day hospital negotiate the presence and involvement of network partners throughout the intervention to create a context of predictable relational availability. Additionally, negotiating the presence and involvement of the clinical team members assists the collaborative network in consistently expressing its care intentions. This involves continuously reflecting on this care intention with network partners and often renegotiating how to express it.

Second, our analysis indicates how the use of networks surrounding the URMs may underscore the continuity of relationships. The collaborative care process negotiates the prolonged involvement of referring partners. It aims to transform them into network partners who, by expressing their continued involvement in the life of the URMs, model a relational continuity that potentially helps to create a safe enough space for the URMs to narrate their suffering and that potentially allows for a meaningful narrative to be built surrounding the proposed care [49].

Third, findings indicate how attempts were made within the collaborative care process to make the URMs’ absent primary caregivers and extended family network present within the collaborative network by proposing them as “present absent” partners [71]. Enquiring about filial duty and about what the family network would think about admission creates possibilities to align the proposed care with maintaining position within the family network. It potentially broadens the theme of relational continuity by creating a perspective in which psychiatric treatment can co-exist with continuity in family relations and in which treatment goals can go beyond symptom reduction towards fulfilling filial duties. By including absent primary caregivers as partners and by envisioning treatment as a means to adhere to filial duty, the collaborative network might come to represent or “become” an extended family network [72].

Fourth, our results show how the collaborative network partners are supported in being more sensitive towards the pervasive impact of both traumatic life experiences and forms of structural inequality. This process enables the collaborative network to become a “circle of witnesses” [39,73] that represents a social community able to acknowledge the symptoms of the URMs as an expression of complex and social suffering more than an individual dysfunction. In line with this, we found that bringing together partners with multidisciplinary backgrounds potentially allows for a more tangible explanation of how semi-residential psychiatric treatment can be socially embedded care that moves beyond a mere medicalization of the URMs’ suffering. Exploring semi-residential psychiatric care together with the URMs as a network of partners evokes practices of collective care that may counteract feelings of social isolation and exclusion and potentially creates a perspective in which psychiatric care can be a means to reconnect with important developmental tasks in resettlement that enable societal integration (language acquisition, academic development, completing asylum procedures, family reunification, etc.). This is in line with holistic approaches to refugee mental health such as the ADAPT model [74] which provides an adaptational framework that identifies the ecosocial context as an important moderator of traumatic events or post-traumatic stressors. In this light, providing pathways towards social embedding and repair is an important aspect of forming a trauma-sensitive response.

Finally, we found that the patient-centeredness that is essential to the practice of collaborative mental health care potentially supports a URM in moving away from a seemingly powerless position towards a more agent one in which the views of the URM on the proposed care become a central aspect of the intervention [75]. We found that the development of this agent position was a continuous effort that required both supportive positioning towards the URMs and continuous dialogue with the network partners. The clinical team repeatedly supported the network partners in both maintaining a position of care when traumatic re-enactment or dysregulation occurred as well as attempting to avoid re-traumatization due to repeating past experiences of coercion and feelings of powerlessness.

However, it is important to note that realizing these dimensions through a collaborative approach comes with a certain degree of complexity and specific pitfalls. Our analysis shows that establishing a perspective of care and appropriately voicing this perspective towards the URMs is a laborious balancing act. Re-enactment of relational and institutional trauma and polarized views on how to understand behavioral symptoms along a “victim–perpetrator continuum” necessitated a careful and complicated management of the collaborative network to avoid splitting between network partners and the URMs or among network partners themselves. Worries and cares of network partners may inform authoritarian and coercive dynamics when met with resistance by the URMs. This creates the requirement of continuous awareness and negotiating of power imbalances between network partners and URMs. Given the fact that these power imbalances are, at least to a certain degree, rooted in social inequalities, it is important to be aware of their persisting nature. Furthermore, the position of care might be perceived as dishonest as the combination of being a network of partners that aims to support the URMs in their decision making and is sensitive towards perceived barriers but supports might feel at times contradictory. This can potentially evoke a sense of untrustworthiness and rejection for the URMs rather than a sense of care. This collaborative intervention should therefore be seen as an attempt to create a “safe enough” space in which cultural differences, socio-structural inequities and feelings concerning a lack of safety can be considered and explored, a space in which these explorations aim to support the URMs in moving towards agency and appropriate decision making together with their surrounding network. Throughout this process, feelings of powerlessness or harm may very well persist [45].

In comparison to interventions such as the Cultural Formulation Interview (CFI) [76] and the Cultural Consultation Service (CCS) [77,78], both aimed at exploring cultural and structural factors impacting illness experience in minority populations, it is relevant to note that these interventions are primarily aimed at informing and optimizing the diagnostic process, while the proposed intervention is aimed at promoting access to mental health care for URMs and does not include diagnostic assessment as such. Another difference between the CFI and the proposed collaborative approach lies in the form of the intervention. While the CFI is centered around a dyadic interaction, CMHC is a systemic approach that focuses on a dialogue between professional partners and patients, their family networks and broader communities. An important difference between CCS and CMHC is that CCS is an intervention primarily focused on, often one-time, expertise-building in cultural and structural competence and cultural safety for professional partners, while the collaborative approach primarily revolves around establishing a working alliance between the patient, their network and professional partners. It does this in a way that makes the patient and their network agentic partners in exploring illness narratives, help-seeking strategies and care needs while simultaneously aiming to target structural determinants and to promote a social network. In this focus, expertise-building of network partners through collaboration with specialized mental health care is integral to mental health care provision.

This study has potential limitations. It is based on qualitative, exploratory research based on cases in which the intervention was completed and successful. The fact that these were successful processes in which barriers were overcome makes the approach at least potentially effective. It does not, however, allow any stringent conclusions to be made on the effectivity of the intervention. Further research both quantitively and qualitatively would be needed to further look into dimensions that inform potential effectiveness. The participants in the study were all unaccompanied, Pashto-speaking, Afghan refugee youth; this impacts the generalizability of our findings for URMs with different socio-cultural backgrounds or for accompanied refugee youth. During the time span of inclusion, four URMs did not complete the intervention. Data on these trajectories were not included in the analysis. Future inclusion of non-successful cases might give further insight into factors that lead to a loss in engagement, while further case-controlled research might provide information about any potential effectivity. Furthermore, due to the reluctance of potential interview candidates, the study only made use of one qualitative interview with a URM.

## 5. Conclusions

Given the substantial mental health needs of URMs and the many barriers they encounter in accessing appropriate mental health care, developing and evaluating interventions that aim to promote the acceptability of and access to specialized mental health care for refugee youth addresses a public health issue. In doing so, it is important to take into account the interplay of barriers originating from cultural and structural determinants of mental health and traumatic suffering.

In line with existing literature on facilitating mental health access for refugees, we propose a collaborative intervention that aims to create a “safe enough” network surrounding the URMs that explores, understands and negotiates relevant cultural, structural and trauma-related barriers, while actively including and aligning to URMs’ views on health and illness, their needs and goals. In detailing the intervention processes and their potential working mechanisms, this analysis equally indicated their complexity.

Future research should look into more detailed role descriptions, a more elaborate exploration of participant experience, different populations and effectivity. This study has several potential implications for further research, practice and policy.

Future qualitative research should look into a more detailed role description of participants such as intercultural mediators and interpreters. Research that includes a broader representation of participants might give more accurate insight into their experience. In particular, a larger inclusion of URMs through qualitative interviews would enable a more detailed understanding of how collaborative mental health care is experienced by URMs and how it might help in making specialized mental health care more acceptable for them.

We would also like to stress the importance of further quantitative research into the effectivity of this intervention. Further qualitative or quantitative research could also look into the use with different populations (e.g., accompanied refugee children).

Our research might be helpful for professionals working with URMs, e.g., people working in the reception network, legal guardians and teachers, in understanding and dealing with the difficulties URMs face in gaining access to appropriate mental health care. It also further stresses the importance of training in frameworks such as cultural and structural competence and trauma-sensitive practice. Furthermore, our findings might illustrate the importance of intersectoral collaboration in working with cultural differences, structural inequities and trauma-related factors and their complex interactions and how the containment of the tensions stemming from this and the prevention of ruptures and splitting in intersectoral partnership can become a potential bearer of trauma repair.

On a policy level, this research might further inform the need for advocacy surrounding the impact of structural factors on mental health and mental health access.

## Figures and Tables

**Table 1 ijerph-21-00292-t001:** Overview of included cases.

Case	Meetings	Network Partners Present	Nationality URM	Language URM	Gender URM	Age URM	Asylum Status
1	1 + 4	Psychologist, counselor and nurse of reception center, counselors of future group home	Afghan	Pashto	m	16	Asylum case pending
2	1 + 3	Legal guardian, counselors of reception center, private psychotherapist, counselors of future group home	Afghan	Pashto	m	16	Asylum case pending
3	1 + 3	Legal guardian, psychologist and social worker of reception center, counselors of future group home	Afghan	Pashto	m	15	Asylum case pending
4	1 + 3	legal guardian, social assistant and adjunct director of reception center, counselors of future group home	Afghan	Pashto	m	16	Asylum case pending
5	1 + 5	legal guardian, counselor, nurse and social worker of reception center, counselors of future group home	Afghan	Pashto	m	16	Asylum case pending
6	1 + 3	legal guardian, counselors of reception center, counselors of future group home	Afghan	Pashto	m	17	Asylum case pending
7	1 + 4	Legal guardian, counselor from current reception center, medical service reception center, counselors of future group home	Afghan	Pashto	m	15	Asylum case pending
8	1 + 5	legal guardian, counselor of reception center, counselors of future group home	Afghan	Pashto	m	17	Asylum case pending
9	1 + 3	Legal guardian, counselor of current group home, counselors of future group home	Afghan	Pashto	m	15	Asylum case pending
10	1 + 3	Legal guardian, counselor of current reception center, counselors of future group home	Afghan	Pashto	m	17	Asylum case pending

**Table 2 ijerph-21-00292-t002:** Overview of participants in qualtitative interviews.

Participants Interviewed for This Study
Clinical psychologist 1 of the day clinic
Clinical Psychologist 2 of the day clinic
Cultural Mediator of the day clinic
Child psychiatrist of the day clinic
Legal guardian of URM 6 and 8
URM 2

**Table 3 ijerph-21-00292-t003:** Overview of all themes and subthemes.

Themes	Subthemes
3.1 Building the capacity to understand and space to contain symptoms of traumatic stress	3.1.1 Creating a holding environment
	3.1.2 Collaboratively exploring symptoms of traumatic stress
	3.1.3 Managing traumatic stress symptoms during the intake meeting
	3.1.4 Collaboratively dealing with traumatic re-enactment
3.2 Shedding light on structural factors impacting the experience of URMs and collaboratively modelling and initiating relative repair	3.2.1 Addressing and containing the impact of migration, asylum and reception policies
	3.2.2 Altering coercive and authoritarian dynamics within the collaborative network
	3.2.3 Collaboratively dealing with institutional distrust
	3.2.4 Constructing a shared therapeutic goal
3.3 Exploring and acknowledging cultural factors, allowing cultural continuity and modelling cultural sensitivity	3.3.1 Welcoming cultural idioms of distress, explanatory models and coping strategies
	3.3.2 Modelling cultural continuity
3.4 Moderating a multi-voiced conversation that explores, gives meaning to, and counteracts stigma	3.4.1 Exploring the meaning of stigma
	3.4.2 Establishing trust
	3.4.3 Redefining psychiatric symptomatology and psychiatric care
3.5 Responding to multiple ruptures and experiences of loss by imagining and enabling continuity	3.5.1 Exploring feelings of ruptures and loss
	3.5.2 Supportive positioning of network partners
	3.5.3 Establishing continuity of family relations
3.6 Forming a network surrounding the URM to counteract loneliness and isolation by embedding care in society	3.6.1 Exploring feelings of loneliness and isolation
	3.6.2 Surrounding URMs with a network
	3.6.3 Envisioning socially embedded care

## Data Availability

The data and materials have not been made available on a permanent third-party archive to protect the anonymity of participants. The data will only be available upon reasonable request to jakob.versteele@upckuleuven.be.

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
