# Peer review of "Developing a Collaborative Approach to Support Access and Acceptability of Mental Health Care for Refugee Youth: An Exploratory Case Study with Young Afghan Refugees"

_ijerph, 2024, doi:10.3390/ijerph21030292_

Round 1
Reviewer 1 Report
Comments and Suggestions for Authors
The manuscript " Developing a collaborative approach to support access and acceptability of mental health care for refugee youth: An exploratory case study with young Afghan refugees." is a qualitative study that provides additional scientific information on a unique way to achieve a ‘safe space’ for a vulnerable group of ten young unaccompanied refugees from Afghanistan, living in Denmark, who were still in various stages of their asylum process. All of them were suffering from various prodromes, probably related to traumatic stress, and were assessed as suitable to benefit from semi-residential psychiatric treatment. The approach includes interplay between cultural and structural determinants of mental health and traumatic suffering. Using within-case analysis and cross-case inductive thematic analysis, the authors identified six central themes that assist access and initiating therapeutic mandate. The study is well documented by the articles cited.
However, there are a few suggestions to improve the manuscript.
The following comments are presented in the order in which they are presented, and not in the order of their importance:
Introduction
1. The introduction presents a detailed, sometimes overly detailed, and up-to-date literature review. To make it easier for the reader, the introduction should be shortened, and chapter heads should be added for each of the topics discussed.
2. The use of the word ‘second’ must be regulated. (p. 2, 82; p. 3, 94)
Materials and Methods
3. “… Level 1 and 2 of Veerman and Van Yperen’s model.” (p.5, 198). The contents must be specified in each of the levels.
4. The authors should refer to those who have not completed the intake process. Doing this will enable the readers to have some idea about for whom, among this specific group of unaccompanied refugee minors (URM), the intervention was not suitable.
This topic should have a specific reference in the Discussion chapter, in addition to relating to the subject in the paragraph of potential limitation (p.15, 738).
5. “A representative of the group home also takes part in one meeting.” [p.5, 225]. It is unclear whether the unaccompanied refugee being interviewed knew the representative/interpreter/clinical psychology/cultural mediator. Please, clarify if any of those being interviewed had a prior acquaintance with any of them.
6. Cases. (p.6, 243-254). It should be explicitly noted that the URMs who participated in the intake process did not master the local language.
Results
7. #3.2.1 The title is messed up. (p.9, 378)
Discussion
8. It would be beneficial to compare the proposed approach to other approaches that relate to the initial phase of the evaluation process (intake) in a cross-cultural mental health context, such as the CFI (e.g. Jervis et al., 2020) or the Cultural Consultation (Kirmayer et al., 2003). In this way, the unique contribution of the discussed approach will also be emphasized.
Kind regards,
Comments on the Quality of English Language
The quality of English writing is very good.
Author Response
In response to the revisions proposed by reviewer one the following changes were made:
Introduction
- The introduction presents a detailed, sometimes overly detailed, and up-to-date literature review. To make it easier for the reader, the introduction should be shortened, and chapter heads should be added for each of the topics discussed.
The introduction was shortened by approximately 16% (or 299 words) to 1571 words and chapter heads (highlighted in text) were added.
- The use of the word ‘second’ must be regulated. (p. 2, 82; p. 3, 94)
The word ‘second’ on p., 82 has been replaced by ‘Next’. As a consequence of this the word ‘second’ now on p.3 , 96 now makes sense as follow up to the Alinea above it marked by the word ‘First’ (p2, 74)
Materials and Methods
- “… Level 1 and 2 of Veerman and Van Yperen’s model.” (p.5, 198). The contents must be specified in each of the levels.
The following sentence has been added: “Level 1 being descriptive levels of evidence detailing the essential elements of the intervention and level 2 being theoretical levels of evidence detailing a rationale based on existing literature.”
- The authors should refer to those who have not completed the intake process. Doing this will enable the readers to have some idea about for whom, among this specific group of unaccompanied refugee minors (URM), the intervention was not suitable.
The sentence “Interventional trajectories for the recruited cases took place during a period spanning from January 2021 through May 2022. During this time span 20 interventional trajectories took place. In these 20 trajectories there were 4 cases in which the URM did not complete the entire intervention.” was added to methodology section.
This topic should have a specific reference in the Discussion chapter, in addition to relating to the subject in the paragraph of potential limitation (p.15, 738).
This specific reference was added to the appropriate section: “During the time span of inclusion 4 URM did not complete the intervention. Data on this was not included. Future inclusion of non-successful cases might give further insight into factors that lead to loss in engagement, while further case controlled research might provide information about any potential effectivity.”
- “A representative of the group home also takes part in one meeting.” [p.5, 225]. It is unclear whether the unaccompanied refugee being interviewed knew the representative/interpreter/clinical psychology/cultural mediator. Please, clarify if any of those being interviewed had a prior acquaintance with any of them.
In this section affiliations of people involved have been clarified by explicitly mentioning their affiliation to the day clinic, the future group home or the network of referring partners:
“The lead in this intervention is taken by a clinical psychologist of the day clinic often accompanied by cultural mediator of the day clinic, the trajectory is supervised by the child psychiatrist of the day clinic who also is present in one consultation. A representative of the future group home also takes part in one meeting. All meetings took place in the presence of an interpreter. The network partners involved in the referral have various professional roles: legal guardians, counselors of reception centers or group homes, ambulatory therapists, representatives of the school,…”
- Cases. (p.6, 243-254). It should be explicitly noted that the URMs who participated in the intake process did not master the local language.
“They were in various stages of Dutch language acquisition, but none of them had mastered the Dutch language to the extent that they would be able to have a mental health encounter in Dutch.” was added to this section.
Results
- #3.2.1 The title is messed up. (p.9, 378)
This was fixed.
Discussion
- It would be beneficial to compare the proposed approach to other approaches that relate to the initial phase of the evaluation process (intake) in a cross-cultural mental health context, such as the CFI (e.g. Jervis et al., 2020) or the Cultural Consultation (Kirmayer et al., 2003). In this way, the unique contribution of the discussed approach will also be emphasized.
An additional paragraph (highlighted in text) was added to the discussion detailing a comparison between CFI, CCS and the proposed collaborative intervention.
Reviewer 2 Report
Comments and Suggestions for Authors
Very interesting manuscript that addresses a very current topic. The mental health care of refugee children and adolescents from a collaborative approach.
The introduction presents in detail the different barriers of the study population in accessing mental health care.
The methodology is appropriate and sufficiently explained, although it would be interesting to know when the study was conducted and a table presenting the team members who have been interviewed.
Results.
In my opinion the results are too extensive, it would be positive to try to reduce them.
I suggest presenting the different topics and subtopics in a table in order to have a clear structure of the results obtained.
I propose that an explanatory paragraph be included for each topic before going on to develop the subtopics it contains.
When each subtopic is explained, an italicized paragraph is included in which verbatim and comments are included. Only the verbatim should be in italics; if the rest of the paragraph is the verbatim of a member of the team, it should be indicated to differentiate between what the researchers say and what the informants say.
The discussion and conclusions are well developed.
Author Response
In response to the proposed revisions of reviewer two the following changes were made:
- The methodology is appropriate and sufficiently explained, although it would be interesting to know when the study was conducted and a table presenting the team members who have been interviewed.
“Cases were recruited during the period of … until… During this time span there were … cases in which URM did not complete the entire intervention due to reasons such as… .” was added to methodology section. A table 2 was added to provide an overview of the interviewed participants.
Results.
- In my opinion the results are too extensive, it would be positive to try to reduce them.
The result section was shortened by approximately 13% (or 561 words).
- I suggest presenting the different topics and subtopics in a table in order to have a clear structure of the results obtained.
A table 3 was added to provide an overview of the themes and subthemes.
- I propose that an explanatory paragraph be included for each topic before going on to develop the subtopics it contains.
Short explanatory paragraphs (highlighted in text) were added to each theme, synthetizing the content of the subthemes found below.
- When each subtopic is explained, an italicized paragraph is included in which verbatim and comments are included. Only the verbatim should be in italics; if the rest of the paragraph is the verbatim of a member of the team, it should be indicated to differentiate between what the researchers say and what the informants say.
Fixed. Only verbatim sections are left in italic. Non-verbatim parts are no longer italicized. All sections referring to actual data are reduced in font and indented relative to the main text.